# Application of Gated Recurrent Unit Neural Network for Flood Extraction from Synthetic Aperture Radar Time Series

**Ming Zhang** [1,2], **Chou Xie** [1,2,*], **Bangsen Tian** [1,2,*], **Yanchen Yang** [1,2], **Yihong Guo** [1,2], **Yu Zhu** [1,2] **and Shuaichen Bian** [1,2]

1   Aerospace Information Research Institute, Chinese Academy of Sciences, Beijing 100000, China; zhangming211@mails.ucas.ac.cn (M.Z.); yangyanchen23@mails.ucas.ac.cn (Y.Y.); guoyihong@aircas.ac.cn (Y.G.); zhuyu@aircas.ac.cn (Y.Z.); bianshuaichen22@mails.ucas.ac.cn (S.B.)
2   University of Chinese Academy of Sciences, Beijing 100000, China
*   Correspondence: xiechou@aircas.ac.cn (C.X.); tianbs@aircas.ac.cn (B.T.)

**Abstract:** Floods are a sudden and influential natural disaster, and synthetic aperture radar (SAR) can image the Earth's surface almost independently of time and weather conditions, making it particularly suitable for extracting flood ranges in time. Platforms such as Google Earth Engine (GEE) can provide a large amount of SAR data and preprocess it, providing powerful assistance for real-time flood monitoring and time series analysis. However, the application of long-term series data combined with recurrent neural networks (RNNs) to monitor floods has been lacking in current research, and the accuracy of flood extraction in open water surfaces remains unsatisfactory. In this study, we proposed a new method of near real-time flood monitoring with a higher accuracy. The method utilizes SAR image time series to establish a gated recurrent unit (GRU) neural network model. This model was used to predict normal flood-free surface conditions. Flood extraction was achieved by comparing and analyzing the actual flood surface conditions with the predicted conditions, using a parameter called Scores. Our method demonstrated significant improvements in accuracy compared to existing algorithms like the OTSU algorithm, Sentinel-1 Dual Polarized Water Index (SDWI) algorithm, and Z-score algorithm. The overall accuracy of our method was 99.20%, which outperformed the Copernicus Emergency Management Service (EMS) map. Importantly, our method exhibited high stability as it allowed for fluctuation within the normal range, enabling the extraction of the complete flood range, especially in open water surfaces. The stability of our method makes it suitable for the flood monitoring of future open-access SAR data, including data from future Sentinel-1 missions.

**Keywords:** recurrent neural network (RNN); gated recurrent unit (GRU); time series analysis; flood disaster; Sentinel-1; synthetic aperture radar

## 1. Introduction

Flooding, one of the most destructive natural disasters worldwide, imposes significant constraints on regional socio-economic development and results in substantial life and property losses annually. The intensity and frequency of precipitation and storms are projected to escalate due to global climate change, heightening the probability of more severe floods that directly impact people's lives [1]. Flood disasters triggered by heavy rainfall exhibit rapid onset and entail widespread consequences. Nevertheless, optical data face limitations based on weather conditions and lack the capacity to effectively monitor floods in a timely manner. As an alternative, employing SAR data, which possesses high availability and operability, proves to be a superior option for flood monitoring [2–9].

The ability of synthetic aperture radar (SAR) sensors to detect floods largely depends on the mirror scattering mechanism of open water surfaces. SAR sensors emit microwave signals towards the Earth's surface in a side view direction, while smooth and open water

surfaces generate mirror reflections—that is, almost a complete reflection of transmitted energy. This reflection results in very low backscatter from the water surface, making the signal received by the sensor very weak [10]. Therefore, many SAR-based flood detection algorithms utilize this scattering mechanism to classify water pixels in a single image by setting a backscatter threshold. These classification methods include global or local histograms [11–17], object-based detection methods [18–24], texture-based detection methods [25–27], region growth-based detection methods [11,28], fuzzy classification-based detection methods [29–32], a method based on the Sentinel-1 Dual Polarized Water Index (SDWI) algorithm [33], and the OTSU algorithm [34–38]. However, the roughness of the water surface is usually high on large water surfaces with strong winds, leading to the presence of capillary waves and causing Bragg scattering. As a result, the backscattering of the water surface is enhanced, which decreases the accuracy of threshold segmentation, especially when using vertical polarization to transmit and receive energy [39].

In recent years, a variety of convolutional neural network (CNN) models have been proposed for flood detection in SAR images [40–46]. Nemni et al. [47] introduced a CNN-based fast flood mapping method, which utilizes a threshold approach to extract water bodies as samples. This method helps to reduce the need for manual intervention, and experimental results indicate that the methods based on U-Net and XNet exhibit the highest accuracy. Another approach, proposed by Konapala et al. [48], combines the U-Net model of CNN with Sentinel-1 and Sentinel-2 data to effectively extract floods. In a different study, He et al. [49] propose a novel CNN model named cross-modal change detection network (CMCDNet), which utilizes both optical and SAR images as inputs for flood detection. This model demonstrates higher accuracy compared to the state-of-the-art method. However, it is worth noting that only Li et al. [50] and Lam et al. [51] have utilized a small amount of multi-temporal data for training CNN models, while almost all other CNN models solely rely on single-temporal data.

With the development of satellite observation technology and the maturity of image preprocessing technology, many satellites are now able to provide long-term and uninterrupted observation data on the ground. For example, Sentinel-1 can provide satellite images with a revisit period of 6–12 days, and some high-latitude areas can even have revisit periods of 3–4 days. Google Earth Engine (GEE) and other platforms can provide a large amount of preprocessed SAR data, offering powerful assistance for real-time flood monitoring [52–55]. These long-term remote sensing data include numerical dimensions, temporal dimensions, and spatial dimensions, forming a four-dimensional information space. Its advantage lies in capturing changes in ground objects at different time points, effectively avoiding the phenomenon of different ground objects having the same backscatter coefficient or the same ground object having different backscatter coefficients [56–59].

By observing the changes in land features over time, we can better distinguish them, thereby improving the accuracy of classification and change detection [60,61]. However, when the numerical fluctuations of time series data are too large, it often leads to significant deviations in the results and a decrease in accuracy [10]. Lu et al. [62] utilized an unsupervised algorithm-level fusion scheme (UAFS-HCD) to extract floods from differential images and improved the traditional hybrid change detection (HCD). Clement et al. [63] used a change detection and thresholding (CDAT) methodology to extract floods from differential interferometric images and created a multi-temporal flood map based on the number of floods per pixel. Amitrano et al. [64] extracted floods based on Haralick texture features and differential image change detection. Cian et al. [65] used new indices such as the Normalized Difference Flood Index (NDFI) and the Normalized Difference Flood in short Vegetation Index (NDFVI) for monitoring floods and shallow water vegetation, using the maximum, minimum, and average values of time series data for index calculations to determine the thresholds for separating flood and non-flood areas. Bangira et al. [66] proposed a new algorithm based on statistical time series modeling of flooded (F) and nonflooded (NF) pixels for near-real-time (NRT) flood monitoring. However, due to the challenges in obtaining and processing such a large amount of data, existing research often

uses a small number of remote sensing images to generate differential images to extract floods and rarely uses a large amount of long-term series data to monitor floods [17].

Time series analyses can reduce the impact of anomalous backscatter features on single temporal SAR images [60,67]. Deep learning has been proven to effectively extract information from large datasets, with little or no need for assumptions about the underlying data and minimal human intervention [68]. Recurrent neural networks, a deep learning architecture, are particularly suitable for processing sequential (such as temporal) data [69,70]. RNNs have been widely used in various fields, including recognition [71,72], classification [73,74], and prediction [75–77].

This study proposes a new gated recurrent unit (GRU) neural network time series SAR image flood range extraction algorithm. This algorithm used images from different years without floods in the previous season as time series data. The Z-scores of the images were used as input parameters to establish an GRU neural network model for predicting normal non-flood conditions. By extracting flood areas based on differences from actual situations, high-precision unsupervised flood extraction was achieved. This algorithm effectively solved the problem of high backscatter values at the center of the water surface and fluctuations in backscatter values caused by image noise. The experiments were conducted in the Dusseldorf area of Germany. The results showed an overall accuracy of 99.20%, with a missed alarm rate of 8.45% and a false alarm rate of 9.78%.

## 2. Site and Event

Located in the center of the lower Rhine River and in the delta area where the Dusseldorf River flows into the Rhine River, the research area is situated in the Dusseldorf region of Germany (Figure 1). Dusseldorf, with a population of 580,000, is an important city in Germany's advertising, clothing, exhibition, and communication industries, as well as a logistics center in Europe. Adjacent to the world-renowned Ruhr district, it is part of the Rhine Ruhr metropolitan area and serves as the capital of North Rhine Westphalia, which is the most densely populated and economically developed region in Europe.

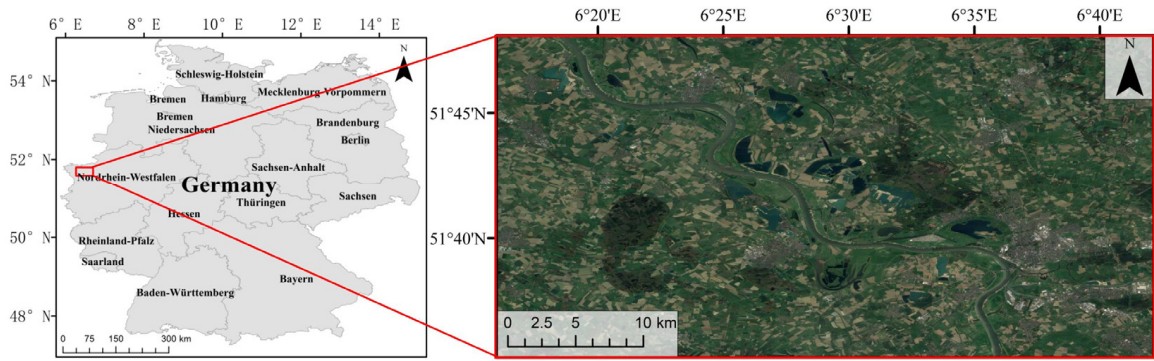

**Figure 1.** Location of the study area with optical image sourced from Google Earth Engine.

The Dusseldorf region is characterized by a temperate marine climate, which features warm winters, cool summers, and minimal annual temperature variations. Precipitation occurs throughout the year, accompanied by overcast skies, foggy conditions, and high humidity. On average, the region receives 797 mm of precipitation annually, with a relatively even distribution across seasons, favoring winter rainfall. The average annual temperature in Dusseldorf is 10.6 °C (51 °F), with minor fluctuations in both daily and yearly terms. The prevailing winds in the area originate from the south or southeast, with speeds measuring between 3 and 4 m (7–9 miles) per second and gusts ranging from 3.5 to 4.8 m (8–10.7 miles) per second. While wind strength in Dusseldorf is generally modest, accounting for approximately 35% of winds below 2 m per second or 4.5 miles per hour, winds are more frequent during winter and nighttime periods [78,79].

Dusseldorf is situated in a region characterized by the deposition of alluvial layers, which consist of mud, sand, clay, and gravel. The terrain rises in the northeast and

southwest directions, while the middle part is relatively lower in altitude (Figure 2). The 10 m land use data provided by Environmental Systems Research Institute (ESRI) Inc. for the year 2021 reveals that the majority of the area is dedicated to farmland, with intermittent urban and forested patches. Moreover, the Rhine River flows through the central part of Dusseldorf, and the surrounding areas along the riverbanks are utilized as grazing lands.

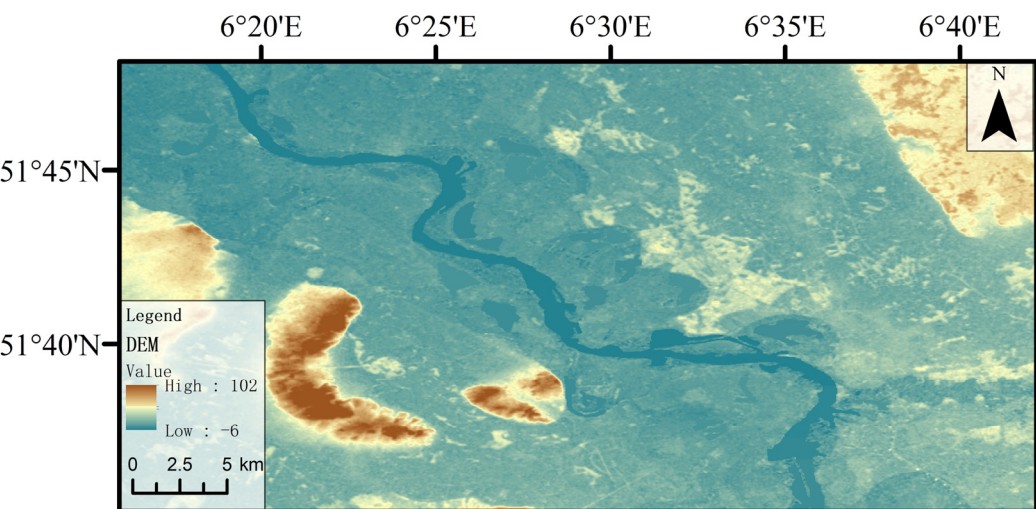

**Figure 2.** Digital elevation model of study area.

In February 2021, a large amount of snow melting and a rainstorm led to floods of varying degrees along the Rhine River in Germany. This event affected nearly 8000 people, 10.5 km of roads, and nearly 4700 km$^2$ of pastures, as reported by Copernicus [80]. Starting from the end of January 2021 and continuing until mid-February 2021, the flood inundated a large number of towns and pastures in the Dusseldorf area. This half-month period caused significant disruptions to people's production and daily lives, resulting in great inconvenience.

## 3. Data and Methods

### 3.1. Sentinel-1 Data

Sentinel-1A and Sentinel-1B, launched in April 2014 and April 2016, respectively, are the first satellites in a series of Earth imaging satellite constellations operated under the European Space Agency's Copernicus program. These satellites collect data in four imaging modes: interference wideband (IW), bar graph (SM), ultra wideband (EW), and waveform (WV) modes. The IW mode is capable of continuously collecting narrow strips of data with an incident angle ranging from 31° to 46°. With two Sentinel-1 satellites orbiting the Earth every 12 days, the IW mode data are available at 6-day intervals over the European region.

In the study area, a total of 242 VV (vertical transmit vertical receive) band data from all Sentinel-1A images collected in IW mode between January 2020 and December 2021 were selected, with a pixel size measuring 10 m × 10 m. Upon inspecting the SAR pixel time series, it was observed that 10 images captured in the VV band, ranging from 31 January 2021 to 15 February 2021, displayed signs of flooding. Notably, the flood on 9 February 2021 exhibited the largest impact range. Consequently, the target image for flood extraction in this study is chosen as Sentinel-1A's VV band orbit reduction image taken on 9 February 2021 (Figure 3). As a comparison, we chose the 23 December 2020 image as the non-flood image (Figure 4).

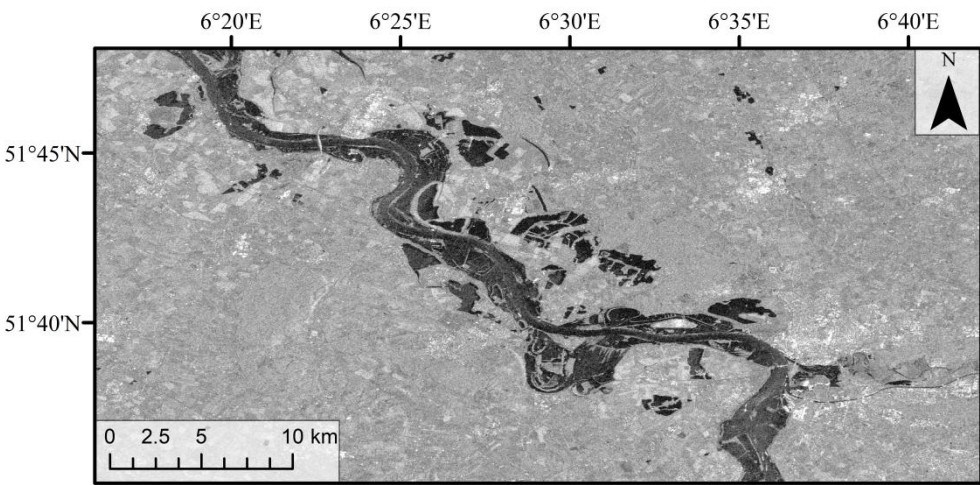

**Figure 3.** The VV band flood image of Sentinel-1A obtained on 9 February 2021.

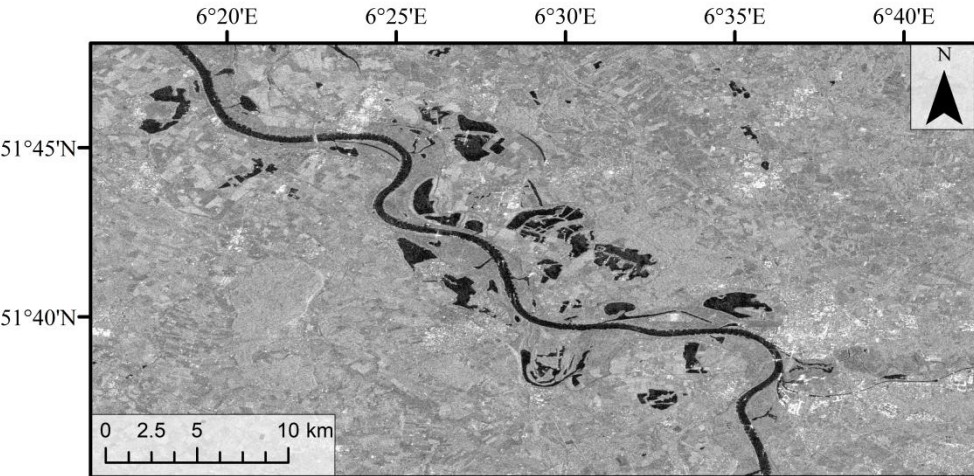

**Figure 4.** The VV band non-flood image of Sentinel-1A obtained on 23 December 2020.

The study selected 18 lowering track images from October 2020 to January 2021 as historical baseline data to predict the normal values of the images on 9 February 2021 if there were no floods. Considering that selecting images with significant time differences or different lifting tracks can have a significant impact on the experimental results, it is important to carefully choose the appropriate dataset. Table 1 provides an overview of all Sentinel-1 data used in this experiment, including VV and vertical transmit horizontal receive (VH) bands.

**Table 1.** Sentinel-1 data used in this study.

| Satelite | Acquisition Date (yyyy/mm/dd) | Polarization | Instrument Mode | Pixel Size (m) | Orbit | Incident Angle (°) |
|---|---|---|---|---|---|---|
| Sentinel-1 A | 2020/09/18 | VV | IW | 10 × 10 | Descending | 35 |
| Sentinel-1 A | 2020/09/18 | VH | IW | 10 × 10 | Descending | 35 |
| Sentinel-1 A | 2020/10/07 | VV | IW | 10 × 10 | Descending | 43 |
| Sentinel-1 A | 2020/10/12 | VV | IW | 10 × 10 | Descending | 35 |
| Sentinel-1 A | 2020/10/19 | VV | IW | 10 × 10 | Descending | 43 |
| Sentinel-1 A | 2020/10/24 | VV | IW | 10 × 10 | Descending | 35 |
| Sentinel-1 A | 2020/11/05 | VV | IW | 10 × 10 | Descending | 35 |
| Sentinel-1 A | 2020/11/12 | VV | IW | 10 × 10 | Descending | 43 |

**Table 1.** *Cont.*

| Satelite | Acquisition Date (yyyy/mm/dd) | Polarization | Instrument Mode | Pixel Size (m) | Orbit | Incident Angle (°) |
|---|---|---|---|---|---|---|
| Sentinel-1 A | 2020/11/17 | VV | IW | $10 \times 10$ | Descending | 35 |
| Sentinel-1 A | 2020/11/24 | VV | IW | $10 \times 10$ | Descending | 43 |
| Sentinel-1 A | 2020/11/29 | VV | IW | $10 \times 10$ | Descending | 35 |
| Sentinel-1 A | 2020/12/11 | VV | IW | $10 \times 10$ | Descending | 35 |
| Sentinel-1 A | 2020/12/18 | VV | IW | $10 \times 10$ | Descending | 43 |
| Sentinel-1 A | 2020/12/23 | VV | IW | $10 \times 10$ | Descending | 35 |
| Sentinel-1 A | 2020/12/30 | VV | IW | $10 \times 10$ | Descending | 43 |
| Sentinel-1 A | 2021/01/04 | VV | IW | $10 \times 10$ | Descending | 35 |
| Sentinel-1 A | 2021/01/11 | VV | IW | $10 \times 10$ | Descending | 43 |
| Sentinel-1 A | 2021/01/16 | VV | IW | $10 \times 10$ | Descending | 35 |
| Sentinel-1 A | 2021/01/23 | VV | IW | $10 \times 10$ | Descending | 43 |
| Sentinel-1 A | 2021/01/28 | VV | IW | $10 \times 10$ | Descending | 35 |
| Sentinel-1 A | 2021/02/09 | VV | IW | $10 \times 10$ | Descending | 35 |
| Sentinel-1 A | 2021/02/09 | VH | IW | $10 \times 10$ | Descending | 35 |

### 3.2. Data Preprocessing

This study employed the European Space Agency (ESA) Sentinel Applications Platform (SNAP) software package for preprocessing. The processing steps are as follows: Firstly, the restituted orbit files were applied to the Ground Range Detected (GRD) image in IW mode to ensure the geometric accuracy of orbit within 10 cm [81]. Next, GRD and thermal noise were removed, which eliminated low-intensity noise and invalid data from the edges of an image swath, as well as reduced additional noise in the subbands and minimized discontinuity between subbands in the image under multi-band acquisition mode in all images [82]. Subsequently, the image underwent radiometric calibration to generate unitless backscatter intensity ($\sigma$) for the image [83]. Next, the image underwent speckle filtering using a Refined Lee filter to remove speckles. For areas above 60° north or south latitude, a Digital Elevation Model (DEM) or Advanced Spaceborne Thermal Emission and Reflection Radiometer (ASTER) Globe Digital Elevation Model (GDEM) from Shuttle Radar Topography Mission (SRTM) was employed for terrain correction and geocoding of these images. Finally, the backscatter intensity was converted to the backscatter coefficient ($\sigma_0$) in decibels (dB) according to Equation (1).

$$\sigma_0 = 10 \log_{10} \sigma \tag{1}$$

This study then collected GRD data using Sentinel-1A in IW mode, which were preprocessed on GEE. For terrain correction and image product output, all images were projected as a WGS84 latitude/longitude (EPSG: 4326) projection.

### 3.3. Statistical Analysis of Time Series SAR Scattering Characteristics

In the time series, the average and standard deviation of the backscatter coefficient were calculated for each pixel in the 18 VV band orbit reduction images separately. Then, Equation (2) was employed to calculate Z-scores, as outlined by DeVries et al. [10]:

$$Z = \frac{\sigma_0 - \overline{\sigma_0}}{\text{std}_{\sigma_0}} \tag{2}$$

Furthermore, for the flood image from 9 February 2021, the aforementioned mean and standard deviation were utilized to compute Z-scores on a per-pixel basis, for the subsequent evaluation of RNN model scores.

*3.4. GRU Neural Network Flood Area Identification*

A RNN is commonly used in continuous data models and has unique advantages in time series classification and prediction. This study referred to Oliver L. Stephenson's [69] GRU, which was chosen because it can learn long-term dependencies in time series, feed back the hidden state output of $H_t$ to the feedforward neural network $H_{t-1}$, and then output the parameters of the predicted distribution. To find the optimal model parameters, the Adam optimizer was used to train the model [69]. Figure 5 demonstrates the mechanism of GRU.

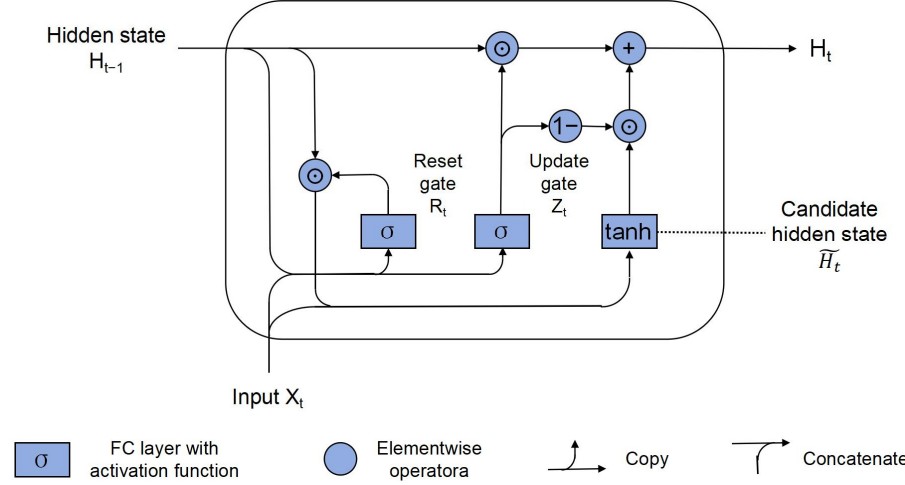

**Figure 5.** GRU schematic diagram.

The reset gate, denoted as $R_t$, controls the retention and storage of previous states. Similarly, the update gate, denoted as $Z_t$, determines the extent to which new states are copies of old states.

The input data for the current time step, $X_t$, and the hidden state from the previous time step, $H_{t-1}$, serve as inputs for both the reset gate, $R_t$, and the update gate, $Z_t$. The outputs of these gates are computed using sigmoid activation functions, σ, and obtained from two fully connected network layers (Equations (3) and (4)).

$$R_t = \sigma(X_t W_{xr} + H_{t-1} W_{hr} + b_r) \tag{3}$$

$$Z_t = \sigma(X_t W_{xz} + H_{t-1} W_{hz} + b_z) \tag{4}$$

where $W_{xr}$, $W_{hr}$, $b_r$, $W_{xz}$, $W_{hz}$, and $b_z$ are the parameters of each gate neuron, which are learned during the training process.

By integrating the reset gate, Rt, with the conventional hidden state update mechanism (Equation (5)) using Equation (6), candidate hidden states for time step t can be obtained. The values of these candidate hidden states are maintained within the range of $(-1, 1)$ using the nonlinear function tanh.

$$H_t = \phi(X_t W_{xh} + H_{t-1} W_{hh} + b_h) \tag{5}$$

$$\widetilde{H_t} = \tan h(X_t W_{xh} + (R_t \odot H_{t-1}) W_{hh} + b_h) \tag{6}$$

where $W_{xh}$, $W_{hh}$, and $b_h$ are the parameters learned during the training process and $\odot$ indicates the element-wise product.

Finally, the effects of the update gate, $Z_t$, need to be considered. The hidden state update formula for GRU is given by Equation (7).

$$H_t = Z_t \odot H_{t-1} + (1 - Z_t) \odot \widetilde{H_t} \tag{7}$$

When the element of the update gate, $Z_t$, approaches 1, the new hidden state is similar to the old hidden state, indicating that the information from the input data, $X_t$, at the current time step can be ignored. Conversely, when the update gate, $Z_t$, approaches 0, the new hidden state, $H_t$, will approximate the candidate hidden state. During the training process, the parameters $W_{xh}$, $W_{hh}$, and $b_h$ are learned to facilitate this integration and element-wise product [84].

This study initially established an GRU model by training the time series of backscatter coefficients of 18 VV band orbit reduction images from October 2020 to January 2021 without flooding as $X_t$. Subsequently, the probability distribution of backscatter coefficients of SAR images under normal conditions without flooding was predicted as $H_t$ using the established model. The image from 9 February 2021 was added to the model, and the mean square error (MSE) was used to calculate the model loss. To determine the final score for damage identification, the consistency difference between the predicted and observed images at the same time was assessed utilizing Equation (8). The predicted average, denoted as $\overline{\mu}$, was compared with the observed value, x, considering the standard deviation of the predicted value, σ:

$$\text{Scores} = \frac{\overline{\mu} - x}{\sigma} \tag{8}$$

We designate areas above 0 as flood areas because they indicate that the actual observed Z-scores are lower than the predicted average Z-scores [69].

To predict the normal values of the image on 9 February 2021, the final consideration is to use Z-score values. For this purpose, the Z-scores of each image from October 2020 to January 2021, which did not experience floods, are calculated using 18 VV band orbit reduction images. The Z-score time series is then used to train the model for predicting the Z-score probability distribution of SAR images under normal conditions without floods. In the next step, the Z-scores of the 9 February 2021 image are added to the model, and the final score is calculated using the same calculation model. If the score of an area is higher than 0, it is considered a flood area.

The study selected a window size of $7 \times 7$ for the analysis. The extracted results were then denoised using a median filtering technique. To account for the presence of objects such as ships and bridges on rivers, which may disrupt the smoothness of the water surface and cause abnormal backscatter coefficients, land use data were incorporated. This allowed for the identification and exclusion of areas with permanent water bodies, thus reducing potential errors in the analysis.

### 3.5. Flood Validation Data

To evaluate the accuracy of the flood map extracted in this study, we used vector data generated by the Copernicus Emergency Management Service (EMS) for the flood event in the Rhine River region on 9 February 2021 as validation data (Figure 6). The pre-event images of these validation data were Sentinel-2A/B images from 17 September 2020 and 19 September 2020. For the flood event images, we used Sentinel-1A images from 9 February 2021 (the same as the images used in this experiment) and COSMO-SkyMed images on 8 February 2021. In order to match the 10 m pixel-sized flood map extracted in this study, we first performed nearest neighbor resampling to rasterize and resample the EMS vector data from Copernicus. We then calculated the missed alarm rate and false alarm rate by overlaying and subtracting the rasterized EMS vector data with the flood grid map we extracted. All the validation work was conducted in Python using the "osgeo/GDAL" and "numpy" packages.

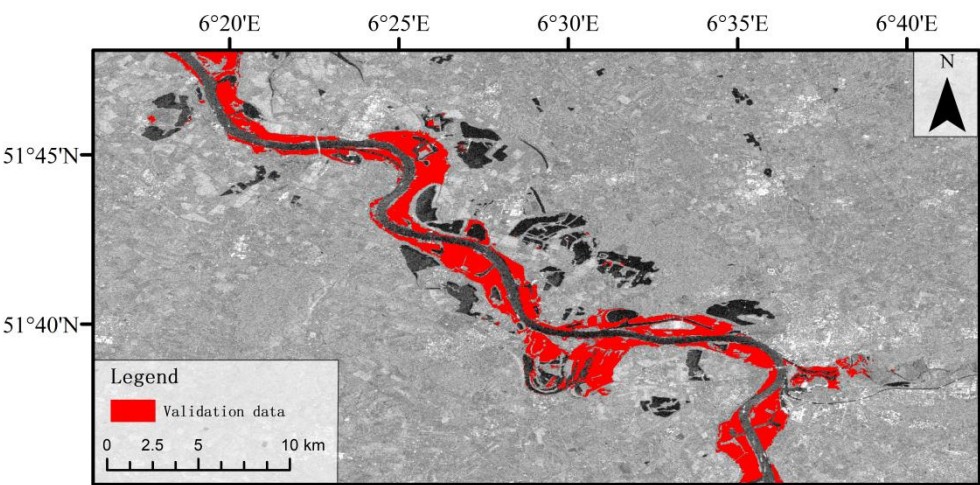

**Figure 6.** Validation data of the study area.

## 4. Results

### 4.1. Evaluation Indicators

The accuracy indicators of the results are defined as the miss rate (FNR), false alarm rate (FPR), and overall accuracy (ACC) in Equations (9)–(11):

$$\text{FNR} = \frac{\text{FN}}{\text{TP} + \text{FN}} \tag{9}$$

$$\text{FPR} = \frac{\text{FP}}{\text{TP} + \text{FP}} \tag{10}$$

$$\text{ACC} = \frac{\text{TP} + \text{TN}}{\text{TP} + \text{FN} + \text{FP} + \text{TN}} \tag{11}$$

where True Positives (TP), False Positives (FP), True Negatives (TN), and False Negatives (FN) are true positives, false positives, true negatives, and false negatives, respectively.

### 4.2. Comparison Method

This study employs the OTSU algorithm, the SDWI algorithm, and Z-score algorithm for comparison.

The OTSU algorithm is a non-parametric and unsupervised automatic method for image segmentation. It is most commonly used to obtain the optimal threshold. This algorithm estimates the appropriate threshold by maximizing the inter-class variance based on the histogram of a bimodal image. The segmentation achieved through this method aims to minimize the probability of misclassification [34].

The SDWI algorithm intensifies the contrast between water bodies and other surface features by multiplying VV and VH polarization images. This enhances water body information significantly. The calculation formula can be seen in Equation (12) [33]:

$$\text{K}_{\text{SDWI}} = \ln(10 \times \text{VV} \times \text{VH}) \tag{12}$$

The Z-score algorithm uses time series data to objectively measure changes in a single pixel [10], enabling rapid differentiation between flood and seasonal inundation regions.

### 4.3. Overall Accuracy Validation

The use of a GRU model to directly perform a time series analysis on backscatter coefficient extracted floods resulted in a missed alarm rate of 0.15% and a false alarm rate of 84.79%, which was too high. The overall accuracy of 75.78% is too low to obtain satisfactory extraction results, and the error is too large. This led to the identification of large areas of

non-flood plain areas and permanent water bodies as floods. Therefore, it is evident that the direct use of GRU for time series analysis is not effective in achieving accurate flood extraction results.

The flood extracted from time series analysis of Z-scores in the VV band of Sentinel-1A at 10 m using GRU demonstrated an overall accuracy of 99.2%, with a missed alarm rate of 8.45% and a false alarm rate of 9.78%.

The flood extracted by using the OTSU algorithm had an overall accuracy of 98.89%, with a missed alarm rate of 15.45% and a false alarm rate of 10.55%.

The flood extracted by using the SDWI algorithm had an overall accuracy of 99.01%, with a missed alarm rate of 16.14% and a false alarm rate of 7.32%.

The flood extracted by using the Z-score algorithm had an overall accuracy of 99.1%, with a missed alarm rate of 15.25% and a false alarm rate of 5.95%.

Missed alarm rate and false alarm rate of different methods are showed in the figures below (Figures 7 and 8).

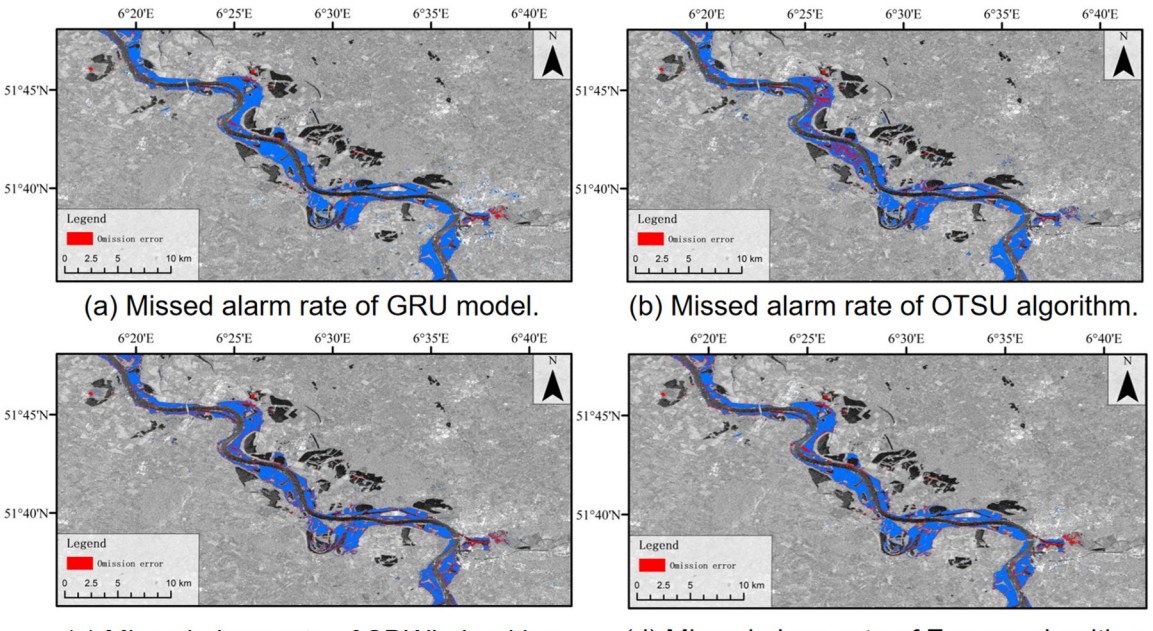

**Figure 7.** Missed alarm rate of different methods.

We found that using Z-scores to establish an GRU model for flood extraction is more effective than directly using the backscatter coefficient for flood extraction, as compared with the accuracy in Table 2.

**Table 2.** Overall accuracy of each method.

|  | Missed Alarm | False Alarm | Accuracy |
|---|---|---|---|
| GRU(Z-scores) | 8.45% | 9.79% | 99.20% |
| OTSU | 15.45% | 10.55% | 98.89% |
| SDWI | 16.14% | 7.32% | 99.01% |
| Z-score | 15.25% | 5.95% | 99.10% |

However, due to the significant difference in the number of flood pixels and non-flood pixels, even if the difference in false alarm rate is significant, the overall accuracy difference is still very small (Table 2). So, considering this situation, we selected two representative local regions to calculate the accuracy.

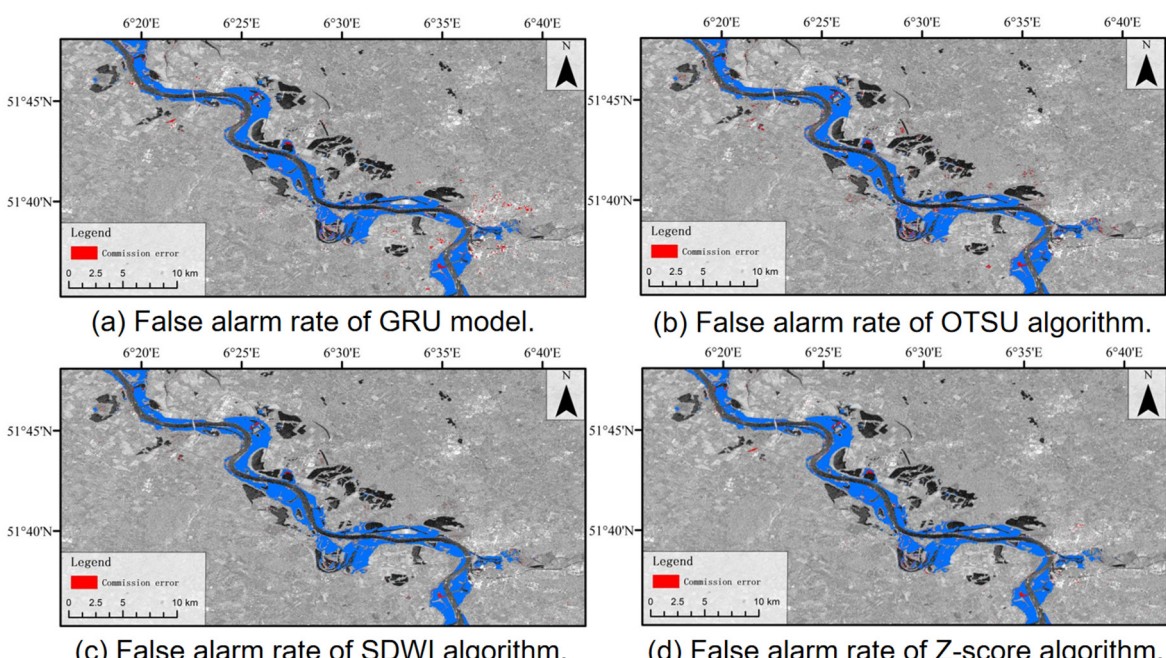

Figure 8. False alarm rate of different methods.

### 4.4. Local Accuracy Validation

We chose local area 1 and local area 2 because there were large areas of open water surfaces and it could clearly demonstrate the different effectiveness of different algorithms in extracting open water surfaces from floods (Figures 9 and 10).

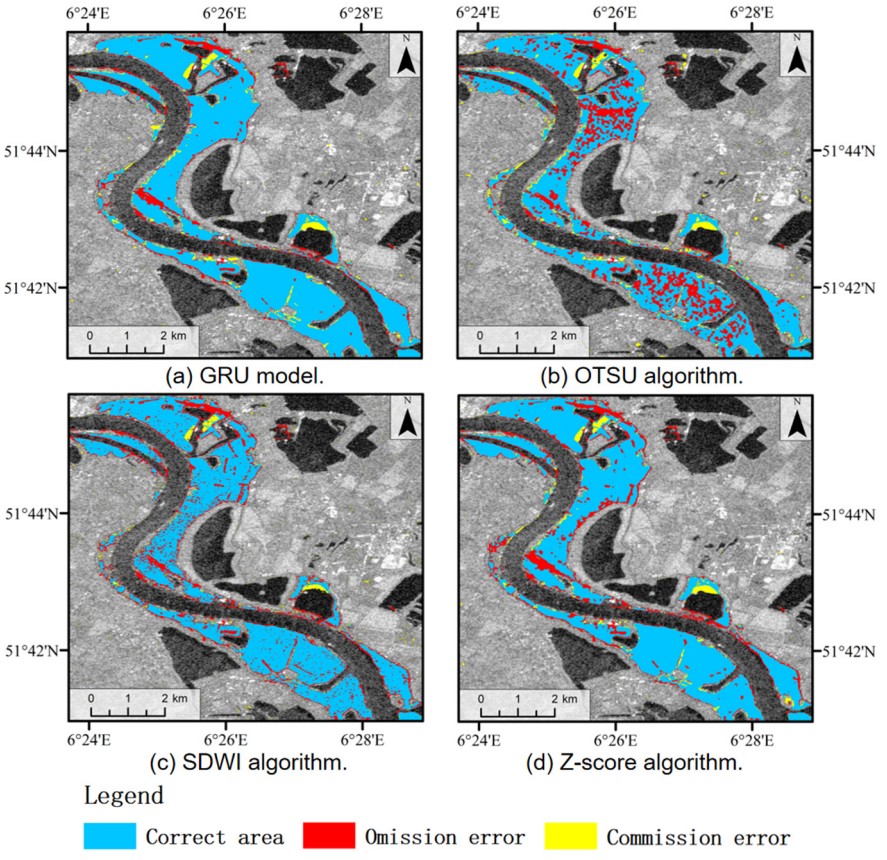

**Figure 9.** Omission and commission error of different methods in local area 1.

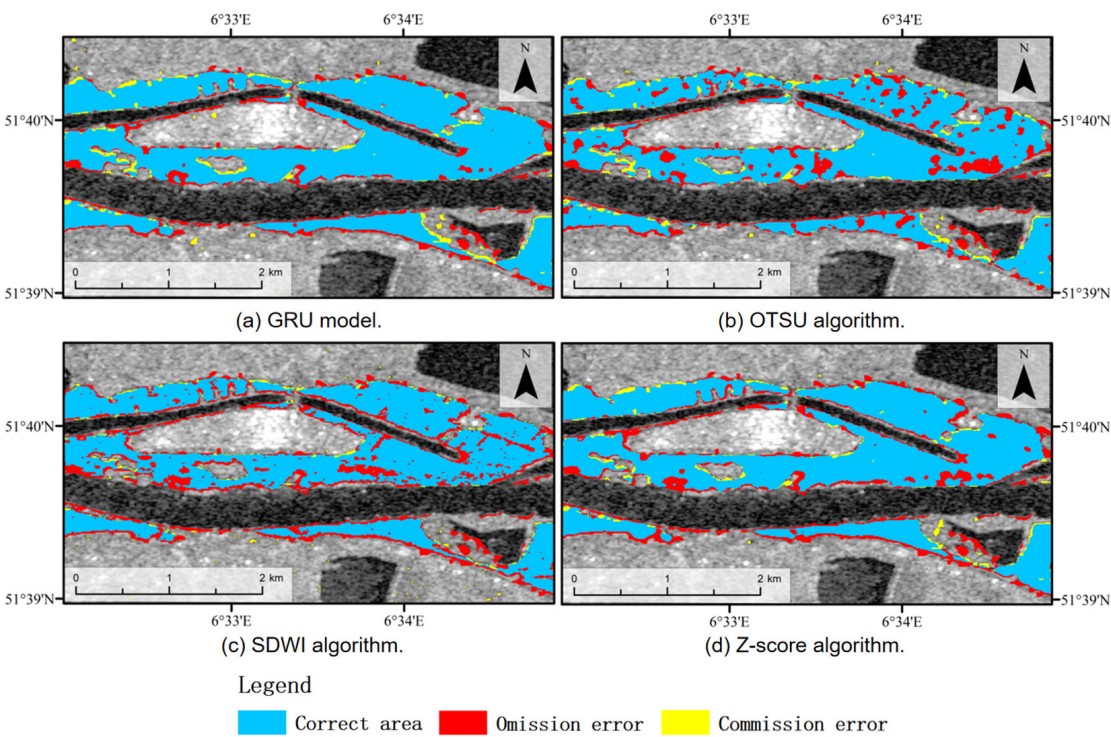

**Figure 10.** Omission and commission error of different methods in local area 2.

The flood extracted from a time series analysis of Z-scores using GRU demonstrated an local accuracy of 97.65%, with a missed alarm rate of 6.84% and a false alarm rate of 7.14% in local area 1. And, the GRU model demonstrated an local accuracy of 94.57%, with a missed alarm rate of 8.99% and a false alarm rate of 6.94% in local area 2. The performance in the center of open water was excellent, as there were almost no isolated pixels. There was a high consistency between the extracted flood data and EMS data, indicating a reliable detection method. Moreover, it could successfully extract the contour of the flood in its entirety.

The OTSU algorithm had an local accuracy of 95.06%, with a missed alarm rate of 22.30% and a false alarm rate of 8.45% in local area 1. And, the OTSU algorithm had an local accuracy of 91.15%, with a missed alarm rate of 19.54% and a false alarm rate of 7.16% in local area 2. It was evident that, even with a window size of $7 \times 7$ applied during the post-processing process, there were noticeable gaps in the center of the open water surface. The median filter of $7 \times 7$, despite being used, still failed to effectively extract the water surface profile of large floods.

The SDWI algorithm had an local accuracy of 96.96%, a missed alarm rate of 12.35%, and a false alarm rate of 6.18% in local area 1. And, the SDWI algorithm had an local accuracy of 92.70%, a missed alarm rate of 18.34%, and a false alarm rate of 3.41% in local area 2. However, there were noticeable gaps in the center of the open water surface and on both sides of the river. Although the SDWI algorithm performed slightly better than the OTSU algorithm in the center of the open water surface, the gap phenomenon was still evident. Furthermore, the extraction effect on both sides of the river and around the lake was suboptimal, resulting in the omission of several areas and a relatively fragmented water body contour.

The Z-score algorithm had an local accuracy of 97.10%, with a missed alarm rate of 13.05% and a false alarm rate of 4.68% in local area 1. And, the Z-score algorithm had an local accuracy of 93.74%, with a missed alarm rate of 14.86% and a false alarm rate of 3.78% in local area 2. Although the Z-score algorithm performed better than the OTSU and SDWI algorithms in the center of open water, it still showed limitations in the extraction performance on both sides of the river and around the lake, where missed alarms were

evident. Directly using Z-scores from a pixel size of 10 m × 10 m was the reason for these limitations.

We found that the accuracy of using the GRU model for flood extraction in open water surfaces was higher than that of using the OTSU algorithm, SDWI algorithm, and Z-score algorithm, presented in Tables 3 and 4.

**Table 3.** Local accuracy of each method in local area 1.

|                | Missed Alarm | False Alarm | Accuracy |
|----------------|--------------|-------------|----------|
| GRU(Z-scores)  | 6.84%        | 7.14%       | 97.65%   |
| OTSU           | 22.30%       | 8.45%       | 95.06%   |
| SDWI           | 12.35%       | 6.18%       | 96.96%   |
| Z-score        | 13.05%       | 4.68%       | 97.10%   |

**Table 4.** Local accuracy of each method in local area 2.

|                | Missed Alarm | False Alarm | Accuracy |
|----------------|--------------|-------------|----------|
| GRU(Z-scores)  | 8.99%        | 6.94%       | 94.57%   |
| OTSU           | 19.54%       | 7.16%       | 91.15%   |
| SDWI           | 18.34%       | 3.41%       | 92.70%   |
| Z-score        | 14.86%       | 3.78%       | 93.74%   |

## 5. Discussion

### 5.1. SAR Image Outlier Detection

This study aims to determine the optimal parameters for SAR image outlier detection in GRU models for flood extraction. Table 2 demonstrates that employing Z-scores for flood extraction yields better results compared to using the backscatter coefficient directly.

From Figures 3 and 4, it can be observed that the transformation of the volume or surface scattering mechanism of the surface from a typical rough land surface (such as rough soil or vegetation cover) to a smooth open water surface leads to a significant decrease in the backscatter coefficient in flood-prone areas. This enables the differentiation between land and water bodies based on changes in the backscatter coefficient. However, the direct use of the backscatter coefficient is not optimal based on experimental findings. This can be attributed to variations in backscattering caused by factors such as incident angle [85,86] or Bragg scattering [87], which complicate the determination of the backscattering coefficient's magnitude. These fluctuations in numerical values have a substantial impact on the backscatter coefficient image, potentially hindering the establishment of the GRU model. Consequently, model convergence is impeded and prediction accuracy is diminished, resulting in a high false alarm rate for the GRU model established using the backscatter coefficient in the experiment.

The backscatter coefficient of each pixel in Z-scores is divided by its own variance in the time series data to reduce the impact of backscatter coefficient fluctuations on the model. This division ensures that only when the backscatter coefficient decreases significantly, i.e., Z-scores $\leq -1$, can flooding be determined. By stabilizing the establishment of the GRU model, the values are allowed to fluctuate within the normal range, which is defined as 1 standard deviation.

### 5.2. Accuracy of Flood Extraction

The GRU model outperforms the OTSU algorithm, SDWI algorithm, and Z-score algorithm in the extraction of floods, as evident from Table 2. It is particularly effective in eliminating the impact of high center roughness on open water surfaces, as observed in Figure 11. This superior performance can be attributed to the GRU model's utilization of time series data and Z-scores. The center of an open water surface is often affected by external factors such as wind, leading to higher roughness compared to smaller water

surfaces. Consequently, the backscatter coefficient is higher and exhibits significant fluctuations across different images. Consequently, the variance calculated using time series data for the pixels within the open water surface increases. As illustrated in Equation (2), higher variance results in a decrease in the absolute value of Z-scores, thereby enhancing the overall stability of the open water surface extraction in the GRU model. This effectively resolves the issue of abnormal backscatter values observed in the open water surface when employing the OTSU algorithm, SDWI algorithm, and Z-score algorithm.

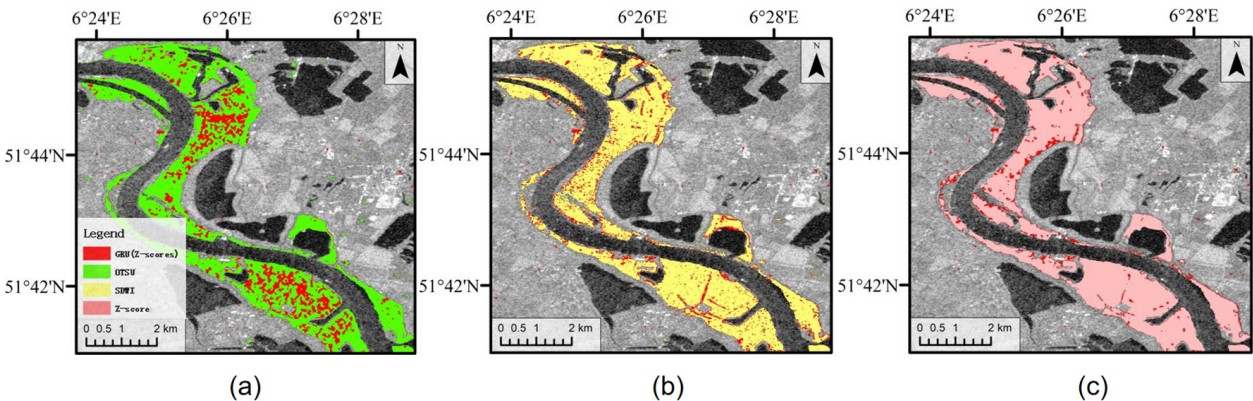

**Figure 11.** The results of the OTSU algorithm, SDWI algorithm, and Z-score algorithm extraction, denoted as (**a**–**c**), respectively, are compared for water surface center extraction.

### 5.3. Strengths of the Method

This study utilized data from previous seasons to establish an GRU model, which offers three distinct advantages:

1. To some extent, the influence of time variations on the fluctuation of the backscatter coefficient within the normal range is eliminated, allowing for fluctuations in the backscatter coefficient within a limited system error, reducing the impact of image noise. By comparing the accuracy of using the backscatter coefficient and Z-scores to establish GRU models, we can see from Figure 12 that Z-scores are more suitable for time series analyses. This is because directly using the backscatter coefficient results in significant fluctuations in time series data, leading to many areas being misclassified as floods. However, Z-scores reduce numerical fluctuations by introducing variance as a divisor, making time series data more stable (Figure 13). Therefore, its high stability provides support for establishing GRU models.

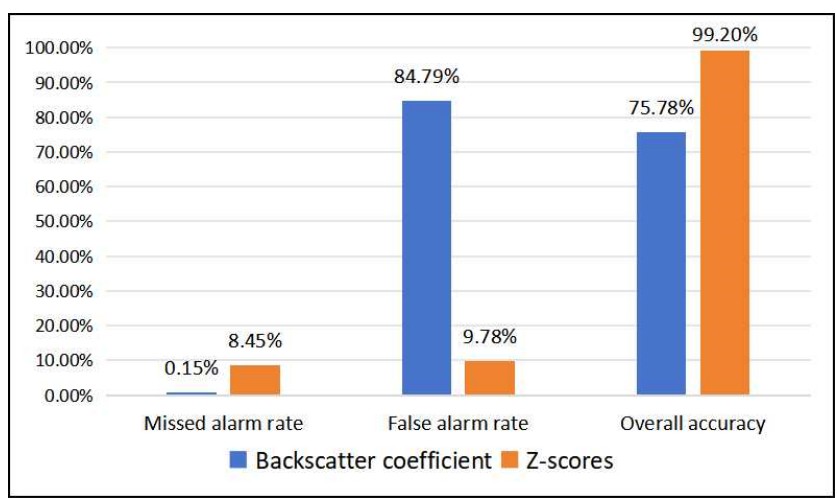

**Figure 12.** Precision of the backscatter coefficient GRU model and Z-score GRU model in non-flood areas.

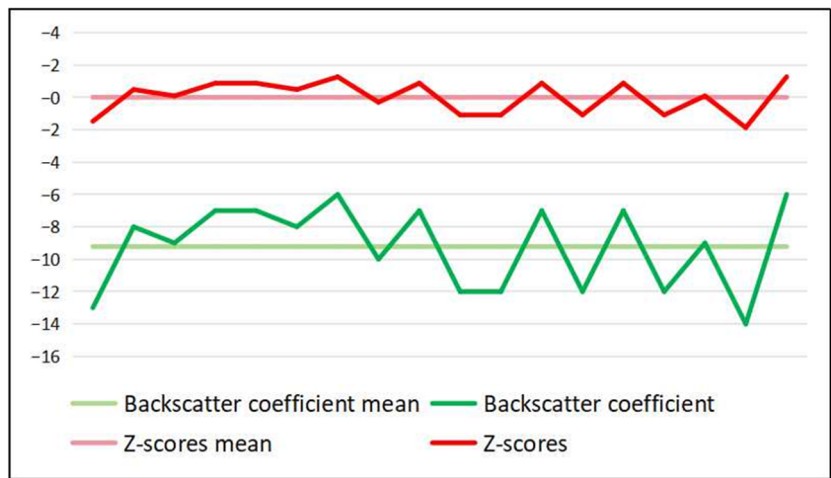

**Figure 13.** Fluctuations of the backscatter coefficient and Z-scores for a single pixel in non-flood areas.

2. In order to effectively address the issue of missed alarms caused by high backscatter values at the center of the water surface, it is necessary to allow for variations in the backscatter coefficient of water within the normal range influenced by wind speed and velocity. Upon observing the extraction results, it becomes evident that the OTSU algorithm exhibits a high false alarm rate, which can be attributed to the fluctuating water level and quality of rivers, lakes, and other areas within the same region throughout different seasons. Additionally, the smoothness of the water surface is affected by factors such as wind power and river flow, consequently impacting the backscatter coefficient of the water and leading to the division of certain areas in the center of the water body into non-flood areas. Furthermore, the OTSU algorithm tends to misclassify areas, such as roads and bare land with smooth surfaces, as water bodies. On the other hand, Z-scores account for the trend of changes (Figure 14) and is capable of differentiating permanent water bodies with minor variations from floods, as well as distinguishing submerged roads from normal roads. Notably, Z-scores do not necessitate the manual selection of additional data from non-flood periods for calculation as it already considers these aspects.

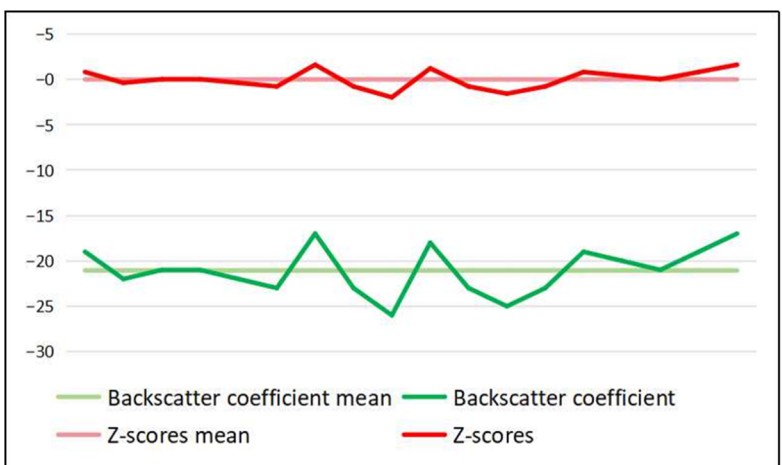

**Figure 14.** Fluctuations of the backscatter coefficient and Z-scores for a single pixel in permanent water bodies.

3. The process of establishing the model does not require manual participation in determining parameters, and the extraction process is fully automated. However, using the GRU model does not require any human intervention, and the threshold is constant at 0. Therefore, simply inputting time series data can automatically obtain the results of flood extraction. Conversely, when extracting floods using the OTSU algorithm, SDWI algorithm,

and Z-score algorithm, it is necessary to manually determine thresholds to distinguish between flood and non-flood areas. This approach has strong subjectivity and low stability and is more labor-intensive and material-intensive.

### 5.4. Limitations and Potential Improvements

First, this method is not applicable in situations where surface changes are too drastic, such as in areas with significant changes like cities, farmland, and seasonal water bodies, which can result in significant errors. Due to the principle of the GRU model detecting sudden changes and extracting floods through relatively stable time series data, significant errors can occur. In the future, if an GRU model that can simulate periodic changes can be established, the impact of normal periodic changes on flood extraction can be ruled out.

Second, we did not consider urban areas and flooded vegetation in this analysis or the validation data (Figure 15). In urban areas and inundated vegetation areas, the double bounce effect can occur due to the vertical and horizontal planes of building walls and vegetation. This effect results in an increase in backscatter values in flooded areas [88–91]. Consequently, using a unified Z-scores division mode makes it difficult to identify these areas, leading to missed detection. In the future, a possible approach could be to use high-resolution images which may help us to eliminate the impact of double bounce effects.

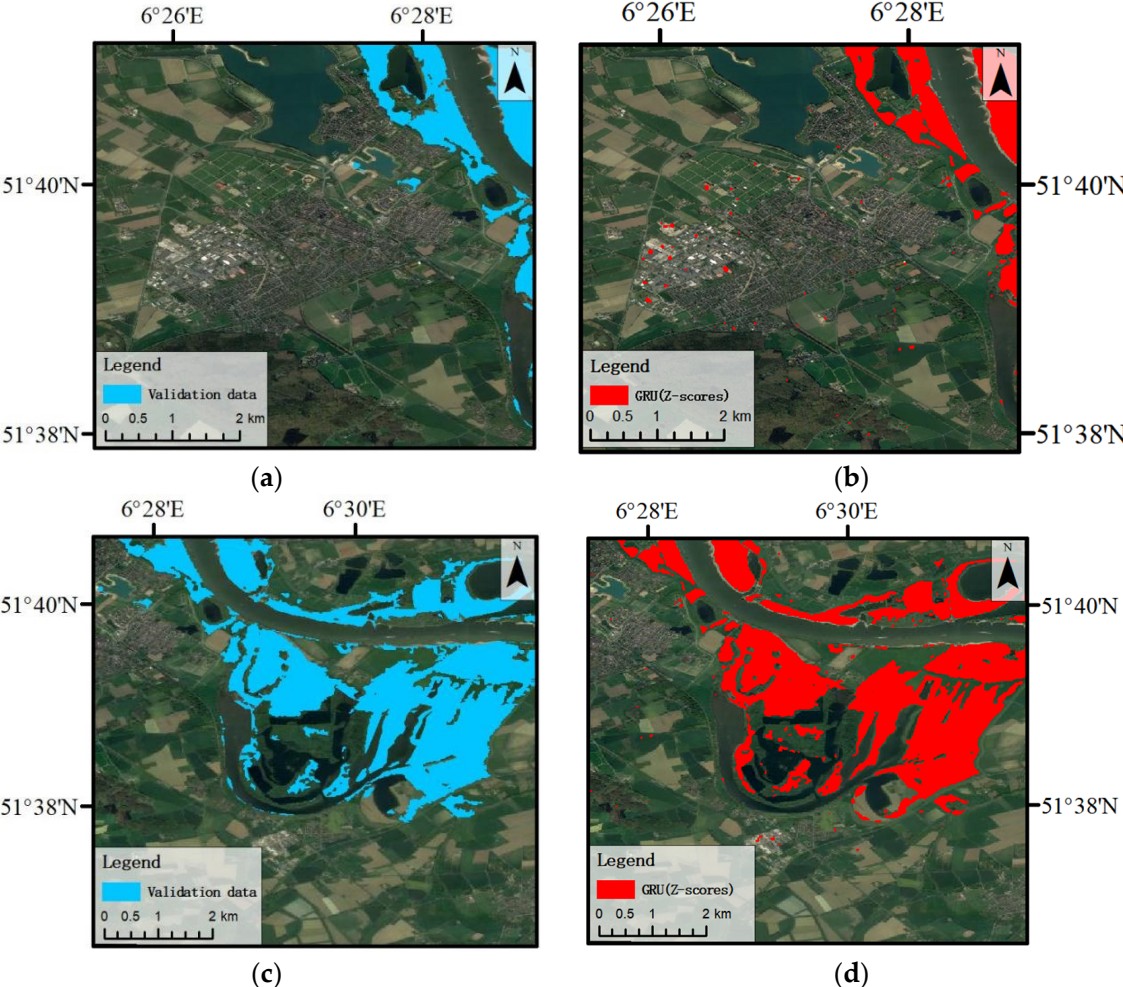

**Figure 15.** In urban areas: (**a**)validation data; (**b**) GRU model (Z-scores). In inundated vegetation areas: (**c**) validation data; (**d**) GRU model (Z-scores).

Third, establishing an accurate time series prediction model requires a large number of continuous images with the same orbital parameters to facilitate real-time monitoring

of surface changes and to establish a more stable model. In addition, floods are a highly dynamic phenomenon, often requiring monitoring of daily or even hourly changes. However, the revisit period of the Sentinel-1 satellite is limited to 6 days, providing data only from the same orbit (Table 2). Although data from other orbits can be added to increase a time resolution, using images with different parameters such as orbits will inevitably introduce errors, resulting in a decrease in the accuracy of flood extraction. Therefore, the accuracy of the algorithm adopted in this study is constrained by the revisit period of the Sentinel-1 satellite.

## 6. Conclusions

In this study, we propose a new method for near real-time flood monitoring, which establishes an GRU model using non-flood SAR image time series, fuses multi-temporal data, and predicts normal non-flood conditions. By comparing the differences between the predicted and actual conditions, we achieve high-precision flood extraction. Compared with the flood maps provided by Copernicus ESM, the method used in our study demonstrates higher accuracy than the common OTSU algorithm, SDWI algorithm, and Z-score algorithm, especially in open water surfaces. This indicates that the new RNN time series SAR image flood inundation range recognition algorithm successfully reduces the impact of open water surface center roughness and image outliers, proving its higher stability and ability to greatly reduce the uncertainty of flood extraction and human factors. In future work, it is essential to establish a GRU model that can accurately predict periodic changes. This will enable accurate predictions to be made in areas with significant seasonal changes. Additionally, separate GRU models can be established for different areas such as cities, inundated vegetation, farmland, grasslands, and forests. Furthermore, data fusion between various satellite images can be performed. The availability of future open access SAR data with higher resolutions and shorter revisit periods, such as future Sentinel-1 missions, TerraSAR, or other commercial satellites, will also be advantageous in improving the accuracy of flood extraction.

**Author Contributions:** Conceptualization, M.Z., C.X. and B.T.; methodology, M.Z. and C.X.; software, M.Z. and Y.Y.; validation, M.Z.; formal analysis, M.Z.; investigation, M.Z.; data curation, M.Z., Y.Y., Y.G., Y.Z. and S.B.; writing—original draft preparation, M.Z.; writing—review and editing, M.Z. and C.X.; supervision, C.X., B.T., Y.G. and Y.Z. All authors have read and agreed to the published version of the manuscript.

**Funding:** This research was funded by the National Key Research and Development Program of China (2022YFC3005601).

**Data Availability Statement:** The data presented in this study are openly available in [FigShare] at [https://doi.org/10.6084/m9.figshare.24438538] (accessed on 23 May 2023).

**Acknowledgments:** The authors thank the editors and the reviewers for their reviews and valuable comments, which significantly improved the quality of this paper.

**Conflicts of Interest:** The authors declare no conflict of interest.

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
