# Peer review of "Application of Gated Recurrent Unit Neural Network for Flood Extraction from Synthetic Aperture Radar Time Series"

_water, doi:10.3390/w15213779_

Round 1
Reviewer 1 Report
1. The Abstract section is very short, it must contains, problem, aim, methodology and the important results of the work.
2. In Site and Event section, authors should add rainfall distribution in the study area, Geology and what about the groundwater setting and recharge.
3. Figure 1 not clear, authors must construct location map for the study area, not from Google Earth Engine.
4. In Data Preprocessing section, something missed in line 190.
5. Figure 4 not clear for the readers.
6. Results section is very short and need more modification.
7. Conclusions section is very short and it must contains recommendations for the future in the study area.
8. References, need to be updated by recent citations.
Reviewer 2 Report
The article topic is an interesting and important one, as early flood detection can obviously help saving a lot of resources. It is also very well structured and presented. However, unless I am absolutely missing something, there must be an error in your accuracy calculation. Maybe a missing term in the denominator? I mean, how can you have FNR 8.45%, FPR 9.78% and ACC 99.2%? Depending on the ratio between TP + FN and FP + TN, ACC has to be somewhere between (100 - 9.78)% and (100 - 8.45)%, right?
Thus your reasoning when comparing the methods is very confusing to me. Judging from the images, GRU displays superior performance in the tested scenarion. Howerver, the accuracy values do not make sense to me, and it is also not clear why accuracy 99.2% means superior performance to 99.1% or 99.01%. Unlike the differences in miss rate, where GRU clearly outperforms the rest. Please clarify this.
---
Some other notes:
It is mentioned that handpicked data were used. How well can it be automated for practical implementation? How time-consuming is data preparation for DNN learning? Is this a problem for practical application?
Line 229: Objective function definition.
How relevant is this score for the real-world scenario? Does better score correlates with better floods warning? This might be more clearly stated in the text.
Figures 6 and 7:
Due to the small size, it is difficult to see FNR and FPR areas. Maybe you could instead try to display all four models in one large image, with different colours.
I do not insist on this one, though, as the results are also nicely displayed in Figure 8 and I understand that showing the information "right" can be sometimes very difficult due to size constraints.
Line 328: Table 2 - "Flase alarm" instead of "False alarm"
Round 2
Reviewer 1 Report
The authors had been modified the manuscript sufficiently, but Figure 1 still not clear. Therefore, the manuscript is accepted for publication in Water MDPI after improving it.